# Validating the CHARMM36m protein force field with LJ-PME reveals altered hydrogen bonding dynamics under elevated pressures

You Xu [1,2,3] & Jing Huang [1,2,3✉]

The pressure-temperature phase diagram is important to our understanding of the physics of biomolecules. Compared to studies on temperature effects, studies of the pressure dependence of protein dynamic are rather limited. Molecular dynamics (MD) simulations with fine-tuned force fields (FFs) offer a powerful tool to explore the influence of thermodynamic conditions on proteins. Here we evaluate the transferability of the CHARMM36m (C36m) protein force field at varied pressures compared with NMR data using ubiquitin as a model protein. The pressure dependences of $J$ couplings for hydrogen bonds and order parameters for internal motion are in good agreement with experiment. We demonstrate that the C36m FF combined with the Lennard-Jones particle-mesh Ewald (LJ-PME) method is suitable for simulations in a wide range of temperature and pressure. As the ubiquitin remains stable up to 2500 bar, we identify the mobility and stability of different hydrogen bonds in response to pressure. Based on those results, C36m is expected to be applied to more proteins in the future to further investigate protein dynamics under elevated pressures.

[1] Key Laboratory of Structural Biology of Zhejiang Province, School of Life Sciences, Westlake University, Hangzhou, Zhejiang, China. [2] Westlake AI Therapeutics Lab, Westlake Laboratory of Life Sciences and Biomedicine, Hangzhou, Zhejiang, China. [3] Institute of Biology, Westlake Institute for Advanced Study, Hangzhou, Zhejiang, China. ✉email: huangjing@westlake.edu.cn

The pressure–temperature ($p$–$T$) phase diagram indicates the thermodynamic range in which the protein structure keeps native or denatured, thus is very informative in protein engineering. While the temperature influence on protein conformational dynamics has been extensively investigated, studies on the protein structure and function at high pressures are relatively rare. Recently the investigations of protein properties concerned with high ambient pressure are boosted in bioengineering, such as enzyme design, food industry and low temperature sterilization[1–5].

Molecular simulations have emerged as important tools for revealing the structure–dynamics–function relationship of proteins. Recently simulations have been applied to not only folded proteins but also intrinsically disordered proteins, as well as proteins under complicated but functionally important environments such as multicomponent membranes[6,7], phase separated states[8,9], and crowded environment in cells[10,11]. The quality of these simulations depends critically on their underlying models, typically the empirical force fields. Protein force fields have been continuously improved; however, the refinement is usually performed under ambient conditions, so that their transferability towards different simulation conditions needs to be scrutinized carefully.

Recent advances in high-pressure instrumentation and the method of collecting samples from the extreme environments at ocean bottom make it possible to study how the enzyme functions are in piezophilic microbes. The fluorescence measurements show that pressure modifies the catalysis constant by affecting the thermodynamic properties thereby was supposed to change the enzyme turnover in engineering[12]. Small angle X-ray and neutron scattering experiments show that the high-pressure stability of calmodulin arises from the reduced void volume inside the protein meanwhile the internal fluctuation is enhanced[13]. The transition states from coiled to folded ubiquitin in declining hydrostatic pressure were captured by nuclear magnetic resonance (NMR) measurements, so the folding pathway was suggested[14]. Indeed the volume change is the key point to capture and evaluate the protein unfolding states[15]. Such study will deepen our understanding of how life origins. Molecular dynamics simulations have been performed at the conditions of high pressures. Unlike temperature, which has a definitely narrow window for protein function, the hydrostatic pressure which causes protein denaturation is varied and such unfolding is usually reversible[15–17]. However, whether current protein force fields which are developed and tested at 1 atm can be used at high pressure such as 1000 or 2000 bar remains an open question.

NMR measurements provide valuable, ensemble averaged information on protein conformations, and therefore have long been used to benchmark protein force fields[18–21]. In particular, through-space scalar coupling $^{h3}J_{NC'}$ detects the strength of protein backbone hydrogen bond (H-bond) N–H···O=C between two residues[22,23] and provides information for both the local H-bond interaction pattern and the global protein conformational dynamics. In 2010 Lange et al. compared ten different protein FFs using the scalar coupling across hydrogen bonds $^{h3}J_{NC'}$ and residual dipolar couplings of ubiquitin and G protein B3 domain[21]. The $^{h3}J_{NC'}$ couplings have also been used in the validation of recent force field development such as CHARMM36m[24] and Amber ff99SB-disp[25]. NMR relaxation is another frequently used measurement to detect the dynamics of residue side chains, as employed frequently in ubiquitin system[26,27]. Such NMR data can be compared with the calculated axis order parameter that is affected by the side-chain flexibility and free volume[28]. A recent NMR study has collected the $^{h3}J_{NC'}$ data for ubiquitin to systematically investigate the protein conformational change in

response to varied ambient pressure from 1 to 2500 bar[29], which provides new data to validate force fields.

LJ-PME represents an important advance in simulation method involving the better treatment of nonbonded Lennard-Jones (LJ) interactions, or more precisely the dispersion (C6) term in LJ potentials. Usually the pairwise LJ interactions are evaluated only within a truncated cutoff radius, for example 9 Å or 12 Å. Although decaying fast with $r^{-6}$, the dispersion term is always negative so that the net sum beyond cutoff radius can be accumulated to an unneglectable value, which is important in highly anisotropic membrane protein systems as well as ligand-protein binding free energy calculations[30]. LJ-PME employs particle-mesh Ewald (PME) algorithm to improve the accuracy of dispersion terms as it has been widely used in electrostatic calculations. It uses Lorentz–Berthelot (LB) combination rules in real space, but the geometric mean in reciprocal space to allow the factorization for the interaction parameters of C6 terms[31]. For common force fields, such as AMBER and CHARMM which employ LB rules for LJ energy calculation, this treatment causes extra computational work while still leaves error in reciprocal space systematically. Wennberg et al. later showed that subtracting an exact term in direct space the errors from reciprocal space can be canceled out, thereby improving the accuracy[32]. With respect to CHARMM force fields, the advantage on calculating long-range dispersion force of LJ-PME in simulating alkane systems has recently been illustrated[33]. However, the performance of LJ-PME with CHARMM protein force field under varied thermodynamic conditions requires validation.

In this work we studied the transferability of the most recent CHARMM36m protein force field (C36m) using NMR experimental data, especially the hydrogen bond $J$ coupling data. We established that the C36m can be readily used together with LJ-PME for protein simulations. Ubiquitin was taken as the model in the MD simulations for its superior thermal stability. We compared the computational $J$ couplings and order parameters with experimental measurements under different temperature and pressure conditions. Furthermore we elaborately analyzed the hydrogen bond features of ubiquitin at elevated pressure up to 2500 bar. The study ends with a brief conclusion on the performance of C36m in thermodynamic transferability.

## Results
### Validations of C36m in varied pressures
*The LJ-PME was feasible with C36m.* As all ubiquitin simulations presented in this study were carried out using the LJ-PME method with OpenMM, we first performed some validation studies. A box of 1890 TIP3P water molecules were simulated in isothermal-isobaric ensemble ($NpT$) ensemble using both switching function and LJ-PME to model the LJ interactions. The density was calculated as the function of pressure (from 1 to 2500 bar) under four different temperatures. As shown in Fig. S1, the results of LJ-PME using 9 Å as cutoff were equivalent to those using 30 Å especially when the pressure is lower than 1000 bar. In contrast, the densities calculated using 12 Å as cutoff were systematically smaller, because the long-range dispersion interaction is not fully accounted. The difference is not significant for water system since electrostatic is the dominant interaction here. In contrast, we recently showed that simulation results of alkanes are more influenced by the treatment of LJ interactions, where with LJ-PME the experimental results were better reproduced[33].

We also validated the ability of the C36m protein force fields in reproducing protein structures and dynamics in combination of the LJ-PME method, by comparing the NMR $^{h3}J_{NC}$ couplings for a set of five proteins. These data were used to validate the CHARMM36 and the CHARMM36m protein force fields with

**Table 1 Correlation between experimental and calculated $^{h3}J_{NC}$ couplings in five proteins.**

| Proteins | Correlation coefficient | | RMSD (Hz) | | Q factor | |
|---|---|---|---|---|---|---|
| | Cutoff | LJ-PME | Cutoff | LJ-PME | Cutoff | LJ-PME |
| 1UBQ | 0.81 ± 0.01 | 0.78 ± 0.01 | 0.109 ± 0.002 | 0.109 ± 0.003 | 0.20 ± 0.00 | 0.21 ± 0.01 |
| 2QMT | 0.78 ± 0.01 | 0.75 ± 0.02 | 0.119 ± 0.001 | 0.128 ± 0.007 | 0.24 ± 0.00 | 0.27 ± 0.02 |
| 1MJC | 0.74 ± 0.01 | 0.76 ± 0.01 | 0.141 ± 0.002 | 0.142 ± 0.003 | 0.27 ± 0.01 | 0.27 ± 0.01 |
| 1QX5 | 0.23 ± 0.02 | 0.33 ± 0.02 | 0.189 ± 0.005 | 0.169 ± 0.004 | 0.43 ± 0.01 | 0.38 ± 0.01 |
| 1IFC | 0.69 ± 0.01 | 0.68 ± 0.01 | 0.163 ± 0.002 | 0.167 ± 0.002 | 0.29 ± 0.00 | 0.29 ± 0.00 |

The 1000 ns MD trajectories were partitioned into ten 100 ns blocks, and the correlation coefficient, RMSD and Q factors for each block were computed. Average correlation coefficients, RMSD and Q factors and their standard errors are obtained from simulations using the C36m protein force field with and without LJ-PME, respectively.

**Table 2 The $p$–$T$ conditions that were adopted in 1UQB simulations.**

| $p$ (bar) / $T$ (K) | 1 | 300 | 500 | 600 | 900 | 1000 | 1200 | 1500 | 2000 | 2500 |
|---|---|---|---|---|---|---|---|---|---|---|
| 278 | 1.69 | | 1.68 | | | 1.76 | | 1.73[a] | 1.72 | 1.64[a] |
| 293 | 1.76 | | 1.69[a] | | | 1.68 | | 1.76 | 1.70 | 1.74 |
| 308 | 1.67 | 1.63 | 1.70[a] | 1.72 | 1.72 | 1.66[a] | 1.72 | 1.69 | 1.72 | 1.73 |
| 323 | 1.72 | | 1.66 | | | 1.74 | | 1.72 | 1.89[a] | 1.73 |

The values show the average RMSD (Å) of heavy-atom coordinates for residues 1–71 calculated over 1.2 μs for each system.
[a]Systems only involved in MD simulations without NMR experiment data[29].

MD simulations run using a 12-Å cutoff for van der Waals (vdW) interactions. As shown in Table 1, very similar correlations between calculated and experimental $^{h3}J_{NC}$ couplings were achieved with the cutoff method and the LJ-PME method for treating the vdW interactions. This is not surprising as for solvated protein systems the electrostatic interactions still dominate the total interaction energy. In comparison, the LJ potential makes more significant contribution to the lipid systems, which hastened the necessity of recent optimization of CHARMM36 lipid FF to explicitly include LJ-PME treatment[34,35]. Even though C36m protein FF was developed using cutoff method, it is still feasible and recommended to directly combine it with LJ-PME method in the simulation of protein systems.

*Structural flexibility of ubiquitin was partially decreased.* During the 1.2 μs simulations of ubiquitin under different pressures and temperatures, all systems reached the equilibrium after first several nanoseconds, and ubiquitin kept in expected stability in varied $p$–$T$ conditions (Fig. S2). The global root mean square deviation (RMSD) of the coordinate for non-hydrogen atoms was evaluated compared to the original 1UBQ conformation. The RMSDs were all below 2 Å and there was no obvious tendency of RMSD change being related to the pressure and temperature (Table 2, Fig. 1). As RMSD of the full structure is insufficient to differentiate the local conformation, the residue root mean square fluctuations (RMSFs) were calculated to get the flexibility of the local domains. The structures, excluding the dangling C-terminal tail, presented similar structural features in all systems: the most mobile subdomain is the hairpin loop (L1) between β1 and β2 strands, whereas the residues in helices and β-sheets were less fluctuated (Fig. S3). Such profile is generally consistent with the thermal motion quantified by the B-factor in crystal structure. The RMSF tendency with respect to pressure is miscellaneous in L1, while in other subdomains the residue fluctuations in high-pressure systems are usually not larger than in low pressure.

We rearranged the RMSF presentation into average values based on the protein secondary structure and residue topology (Fig. 2). The trend of temperature effect is clear. Despite higher temperature generally caused larger fluctuation, the largest RMSF difference between 278 and 323 K is 0.3 Å, which reproduces the strong thermal stability of ubiquitin. On the other hand, the pressure influence is ambiguous. In different temperature groups, the trends of pressure influence were not consistent, for instance, the linearity of correlation between RMSF and pressure in 293 and 308 K is better than that in 278 and 323 K. The average RMSF in α-helices is mildly reduced with the pressure ranging from 1 to 2500 bar, whereas in β-sheets and loops such trend was only observed in 293 K (Table S1). For backbone atoms both helices and sheets have lower RMSF than loops, and for side chain atoms sheets and loops have higher RMSF than helices. In summary the local rigidity of ubiquitin structure is ranked as helix > sheet > loop, and the fluctuation reduction in response to pressure is, helix > sheet and loop.

*Backbone H-bond dynamics reproduced using $^{h3}J_{NC'}$ couplings.* At least 53 backbone hydrogen bonds were observed in all systems and 37 of them were very stable in the simulations (Table S2). To validate the performance of C36m in different thermodynamic conditions, the $J$-coupling across atoms N–H···O=C ($^{h3}J_{NC'}$) was calculated from MD simulations and compared with the measurements of 33 backbone H-bonds in NMR study[29]. The discrepancy of calculation from experimental value, $\Delta^{h3}J_{NC'} = {}^{h3}J_{NC',\ calc} - {}^{h3}J_{NC',exp}$, was adopted to evaluate the agreement between MD simulations and experimental data. For each system the average of absolute differences $\Delta^{h3}J_{NC'}$ for measured H-bonds is lower than 0.1 Hz, which suggests the productions were consistent with experiment (Fig. 3a). Taking account of the fact that many NMR detected $J$ couplings are smaller than 0.5 Hz, we took the experimental value as denominator so the averages of relative discrepancy are 17–27%. As a general view, no evidence shows the systematic deviation in any specific temperature group, nor is the discrepancy going higher as pressure increasing. This indicates the C36m FF, which is parametrized at room temperature and pressure, performs equally well for protein simulations under the varied thermodynamic conditions in range of 278–323 K and 1–2500 bar.

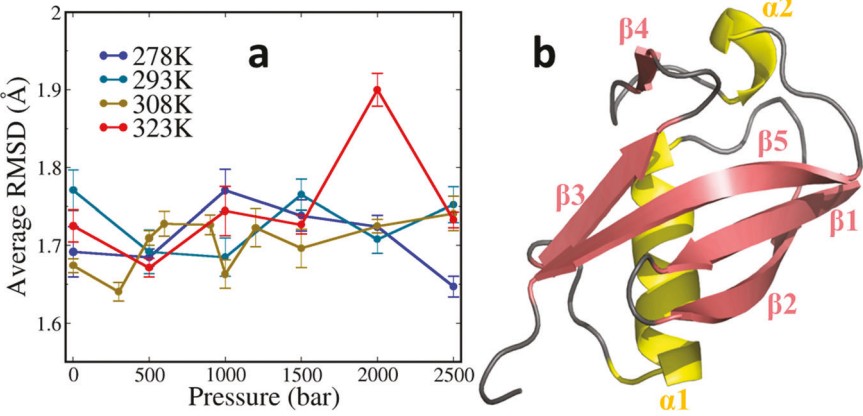

**Fig. 1 The overview of the simulation. a** RMSDs from crystal structure of heavy-atom coordinates calculated in each system. Each value was averaged from the last 1 μs simulation with the block standard deviation shown as the error bar. **b** The 3-D structure of 1UBQ with yellow for helices, pink for sheets and gray for loops.

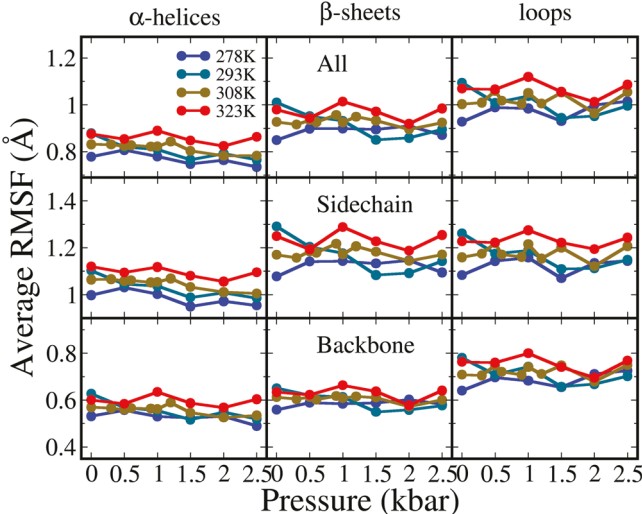

**Fig. 2 The average RMSF of heavy atoms in all systems.** Three rows show the data of all atoms, side chain atoms and backbone atoms, respectively, and three columns for helices, sheets and loops, respectively.

Most discrepancies are distributed equally close to zero, but significant deviations were observed for a number of H-bonds which contributed significantly on the average results. Setting ±0.1 Hz as the upper and lower bounds, the H-bond strengths of K6-L67, R42-V70, L67-F4, K33-K29, and I61-L56 were mostly overestimated, while T7-K11 and L56-D21 were underestimated (Fig. 3b). The reproduction of these H-bonds has a systematic discrepancy to NMR measurement in all systems. On the other hand, compared to the crystal structure 1UQB all the H-bonds were well reproduced and stable in each system, indicated by H···O distances shorter than 2.4 Å with small fluctuations (Fig. S4). The weak H-bond S65-Q62 (not presented in NMR study) was also maintained. The hydrogen bonds in L1 domain indeed breathed more frequently than other domains, in accord with the high residue RMSFs.

Although these H-bonds were very stable in simulations and the distances at high pressures were generally shorter than at low pressures, the relationship between the interaction strength and pressure is not straightforward. The Pearson correlation coefficient ($\rho$) was adopted to evaluate the linear correlation between H-bond strength and pressure. It indicates the linearity but not the value of slope. As shown in Fig. 4, the H-bonds with positive correlation in three or four temperature groups are more than those with negative correlation, which indicates the interactions of most H-bonds were likely enhanced from the elevated pressure.

The H-bonds that had positive pressure correlation in all temperatures include F4-S65, V17-M1, D21-E18 I23-R54, E34-I30, L56-D21, S57-P19, I61-L56, E64-Q2, S65-Q62, and H68-I44. Those correlations are consistent with the sign of derivative $\delta|^{h3}J_{NC'}|/\delta p$ reported in the NMR study[29], except for D21-E18 and S65-Q62 which were not detected experimentally probably due to the weak signal. In contrast, only two cases M1-V17 and R72-Q40 had fully negative correlations. They are the first and last backbone H-bonds which gate the β1/β2 sheet (N-terminal) and β3/β5 sheet (C-terminal) respectively. Note that the two H-bonds between residue M1 and V17 had opposite responses to pressure. This observation resembles the case of R42 and V70 reported in NMR study, where V70-R42 has positive correlation whereas R42-V70 is the opposite (Fig. S5).

The H-bonds I3-L15, K6-L67, A28-E24, K48-F45, L50-L43, and N60-S57 had negative response with pressure in at least three temperatures. Their derivatives $\delta|^{h3}J_{NC'}|/\delta p$ however are not significant, suggesting the pressure destabilization is not significant. Except for A28-E24, K48-F45, and N60-S57 which have not been reported experimentally, the signs of their derivatives are consistent with the NMR study. The similar consistency was also maintained for the H-bonds with $\rho > 0$ in three temperatures. A considerable contrast is L69-K6, whose positive linear correlation in experiment was not reproduced in the simulation. The remaining H-bonds have either random signs of $\rho$ or low $|\rho|$, and display no linear correlation in this study. Those H-bonds may not be influenced by current thermodynamic conditions.

*Methyl motions were partially restricted.* The internal motions of protein measured by NMR relaxation often respond differently to the system thermodynamic conditions. To evaluate the effect of pressure on the methyl-bearing motion of the ubiquitin, the order parameters ($S^2$) of side chain methyl group were calculated. Two data sets of NMR $^2$H relaxation at 303 K under 1 atm[36] and various ambient pressures (1, 400, 800, 1200, 1600, and 2500 bar)[26] were taken as the references. Both experiments show the similar profile of $S^2$ distribution yet some values are different. The methyl groups with low $S^2$ bear more motion in global conformational equilibration. The reproduced order parameters for all methyl axes of side chain are basically consistent with the profile of NMR data (Fig. 5). Such distribution of internal motion for methyl groups has some similarity with residue RMSF, where

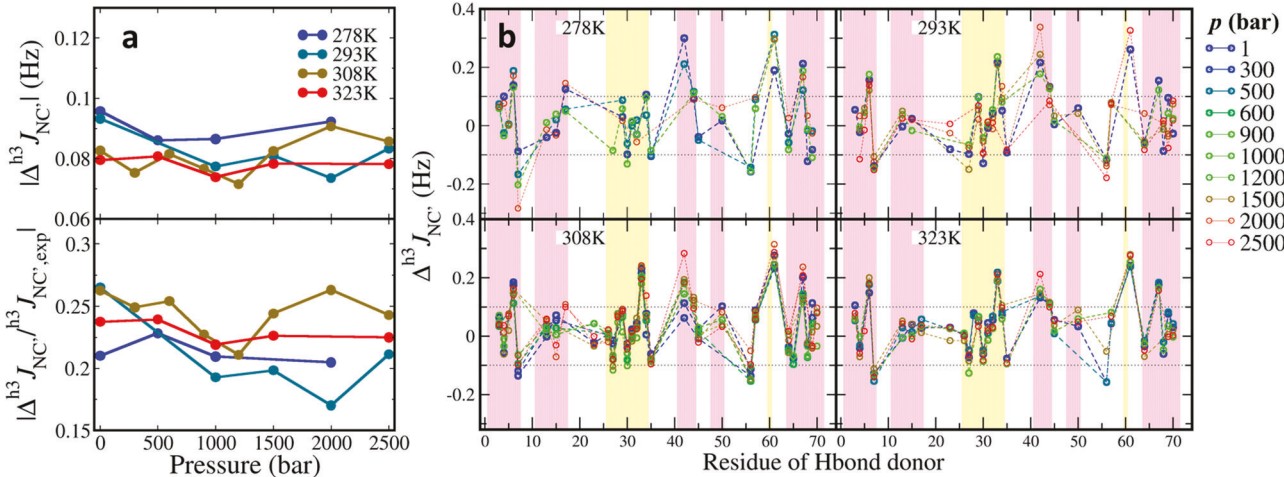

**Fig. 3 The discrepancy of calculated $^{h3}J_{NC'}$ coupling from the experiments. a** The absolute average $^{h3}J_{NC'}$ difference of each system. The upper panel shows the average of absolute difference and the lower panel shows the ratio of absolute difference over experimental value. **b** The residue-wise $\Delta^{h3}J_{NC'}$ of all systems. Each panel organizes the $\Delta^{h3}J_{NC'}$ of different pressures in a specific temperature. The hydrogen bonds are indexed by the donor residue. The panel background is colored by the secondary structure domains, *i.e.*, yellow for helices and pink for sheets. The dotted baselines indicate the thresholds ±0.1 Hz.

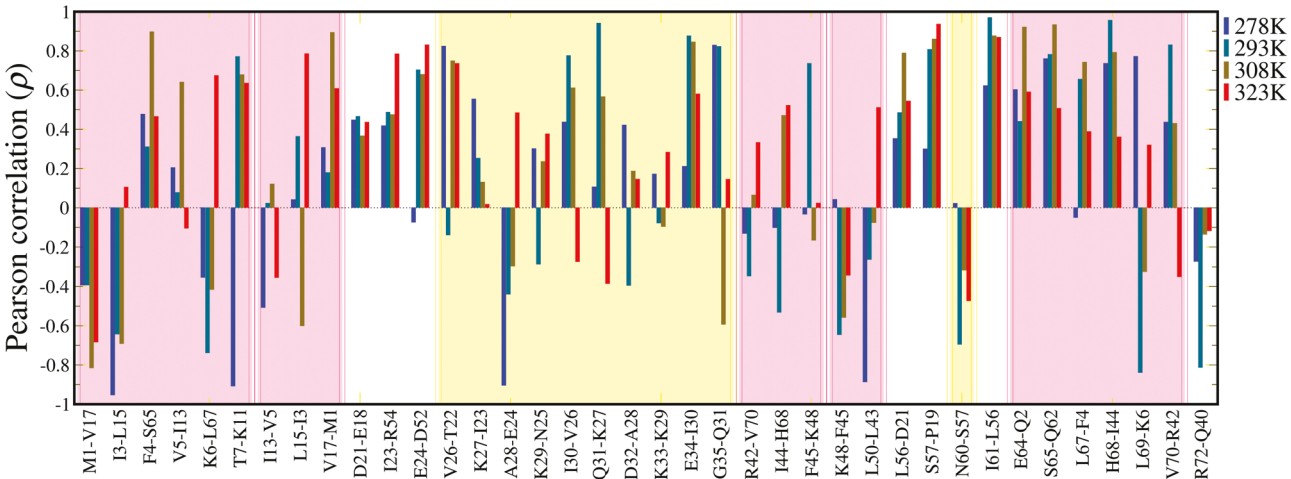

**Fig. 4 Pearson correlation between the $^{h3}J_{NC'}$ of backbone H-bond and hydrostatic pressure.** The dashed line at $\rho = 0$ indicates no linear correlation at all, while $\rho = 1$ and $\rho = -1$ indicate totally positive and negative linear correlation respectively. The background is colored by the secondary structure domains, i.e., yellow for helices and pink for sheets.

L1 and β3 were more mobile than others. The underestimation in some sites, such as T7γ2, T9γ2, and I30δ, was also observed in previous simulations of CHARMM36 in room conditions[20], but the error magnitude in this study using C36m is smaller.

The NMR measurements indicate many methyl groups have increased $S^2$ values in response to the ambient pressure. For residue of low order parameters, they are usually more sensitive to this effect. Those groups may be attached to both solvent exposed residues, for instance L8δ1/2 and I44δ, and buried residues such as L67δ1/2. The calculated order parameters increased at elevated pressure as well, for most residues especially in β-sheets and loops. Some residues even over-responded to the pressure so that their internal motions reduced more significantly than the experimental measurements. Such overestimation of pressure effect however brings the magnitude of calculated $S^2$ in high-pressure closer to NMR data, for instance, the methyl groups of I3γ2/δ, T7γ2, I13δ and I30δ. Sites that did not reproduce the consistent pressure effects include I13γ2, L67δ1/2 and V70γ1/2, where the calculated trends are opposite to NMR

observations. For the residues of intrinsically less mobility ($S^2 > 0.8$), especially those located in the α-helix, the order parameters were not sensitive to the ambient pressure, in agreement with experiment.

**Ubiquitin transition states in response to pressure**
*Pocket was compressed but not penetrated by water.* From 1 to 2500 bar ubiquitin structure bore a compression, however the conformational change of most local domains was subtle in simulations. Ubiquitin is a typical globule protein in which most hydrophilic residues are solvent accessible and most hydrophobic residues are buried and thus solvent inaccessible. We evaluated the deformation of ubiquitin inner part to check how elevated pressure influences its structure.

Among 28 hydrophobic amino acids in the sequence of ubiquitin 14 residues whose side chains aggregated inward from the solvent accessible surface were identified to be the hydrophobic core. We monitored the distances between the centers of mass of those 14 hydrophobic side chains, and calculated the

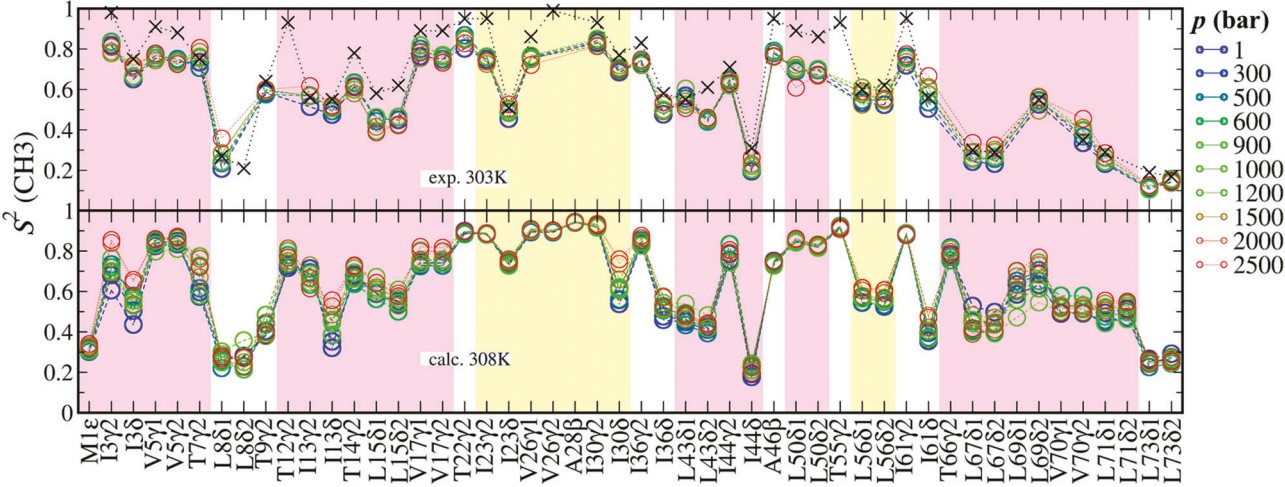

**Fig. 5 Order parameter ($S^2$) of side chain methyl group in ubiquitin.** The x-axis is indexed by the methyl carbon name of the residues. The upper panel shows the data from two experiments (circles under varied ambient pressures[26] and crosses at 1 atm[36]) at 303 K and the lower shows calculated data at 308 K. The background is colored by the secondary structure domains, i.e., yellow for helices and pink for sheets.

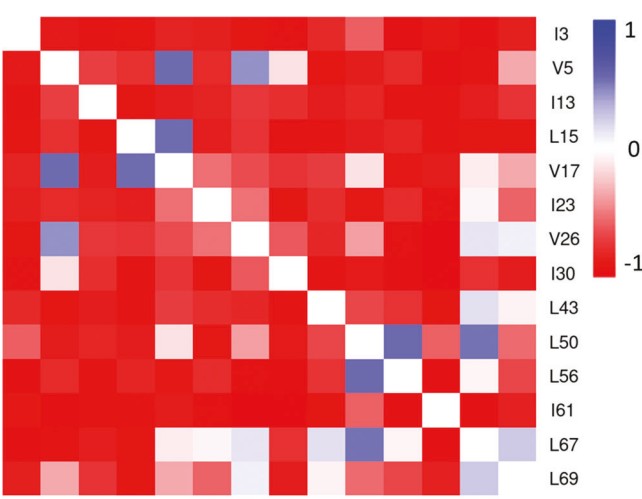

**Fig. 6 Pearson correlation coefficients between hydrophobic distances and pressure at 308 K.** The pairwise distances were measured between the mass centers of 14 hydrophobic side chains in the pocket. From $\rho = 1$ (blue) to $\rho = -1$ (red), the distances correlate with pressure from positively to negatively.

Pearson correlation between each distance change and the ambient pressures. The map of distance-pressure correlation from simulations at 308 K is shown in Fig. 6, and the results of the other three temperatures are in Fig. S6. Despite the overall changes were subtle, their coefficients mostly indicated negatively linear correlations. This suggests the pairwise distances between side chains in the hydrophobic core commonly were decreased linearly with pressure in the compression. Some cases that had weak correlations ($0.5 < \rho < 0.5$) indicate these distances were not homogenously changed, which suggests that the pressure effect on protein shape might not be isotropic and the dimension of the hydrophobic core was mildly adjusted in response to the compression. The cavity volume of ubiquitin was also evaluated for the systems at 308 K using CAVER3.0[37] (Fig. S7). Tunnels were vanished gradually as bottleneck radius was decreased, which confirms that the cavities in ubiquitin are compacted by pressure.

Since those residues aggregated into a compact form, they bore very low RMSF compared with more exposed residues (Fig. S3). The space of compressible pocket volume therefore is very limited, and this is remedied by reducing the vdW radii of these residues. The smooth reduction shown in the coefficient matrix also implies that no water molecule had been passed through those hydrophobic groups. This suggests that in the simulations, ubiquitin will not be destabilized by water penetration as pressure is up to 2500 bar.

*Global hydrogen bonds had different features.* Additional to the H-bonds between backbones, donors/acceptors of side chain were also involved in hydrogen bonds and contribute to the structural stability of ubiquitin. Due to bigger mobility, more than 100 side chain involved H-bonds were monitored in simulations, but most were transient based on the average effective occupancies over all systems. While 37 of 53 backbone H-bonds had more than 70% occupancy, the side-chain H-bonds were much weaker as only 21 cases were lasting more than 30% simulation time (Table S2).

Ubiquitin is abundant of charged residues where 11 bases and 11 acids from 41 hydrophilic residues are distributed in whole domains. Their functional groups participate in all the side chain —side chain and half of the backbone—side chain H-bonds in MD simulations, thereby bringing additional electrostatic interaction to those H-bonds. There are 5 salt bridges (H-bond between a base and an acid residue) in the crystal structure. During simulations their average occupancies were ordered as: K27-D52 (0.95), K11-E34 (0.59), K63-E64 (0.36), R54-E51 (0.28), and R72-D39 (0.21). While lysines kept the H-bonds stable, the arginines were more flexible and their original H-bonds were rotated to R54-D58 (0.58) and R74-D39 (0.33) respectively. Despite of the mobility, salt bridges are still the most stable H-bonds among all side chain ones.

For the stable side chain H-bonds (occupancy > 30%), the Pearson correlation coefficients between occupancy and pressure were presented in Fig. 7a. The most stable K27-D52 was less affected by pressure, while the other salt bridges had almost negative correlation with pressure. The exception R54-D58 was interconverted with R54-E51. Since R54 is located between E51 and D58, the correlations of these two H-bonds are opposite. At lower temperature, R54-E51 was enhanced by pressure but at high temperature it turned to enhance R54-D58. Besides salt bridges, the remaining H-bonds between side chains with

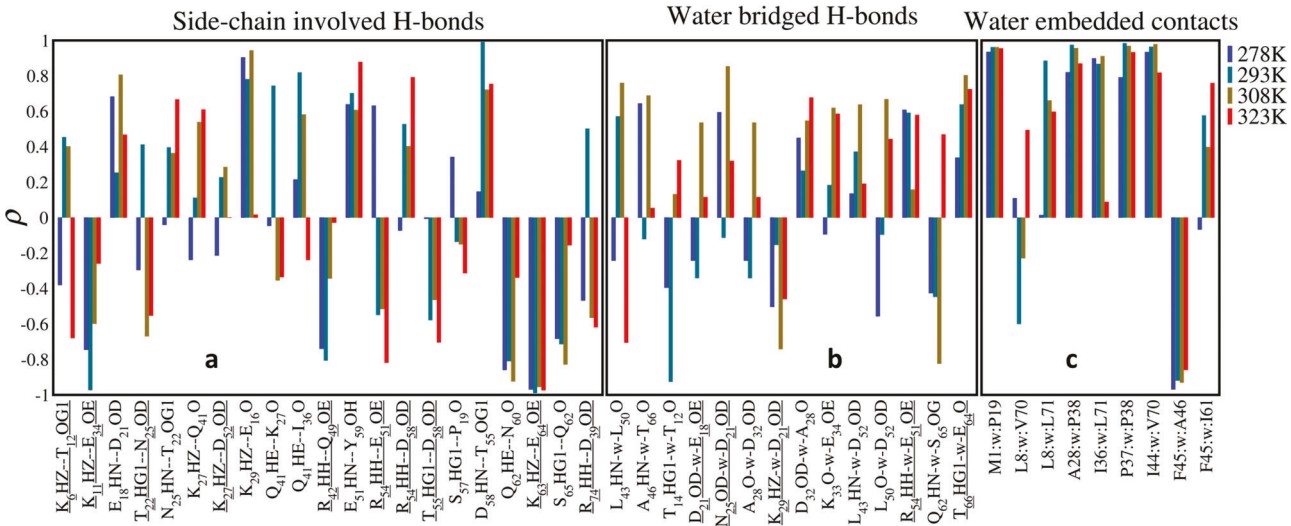

**Fig. 7 Pearson correlation between the effective occupancy and pressure for hydrogen bonds or hydrophobic contacts. a** The donors/acceptors of side-chain involved direct H- bonds. **b** The water-bridged H-bonds. **c** The water-bridged hydrophobic contacts. For (**a**, **b**), the x-axis labels both specific residue and atom involved in the H-bonds, and the ones with underline indicate the H-bonds between side chains and the remainings are between backbone and side chain.

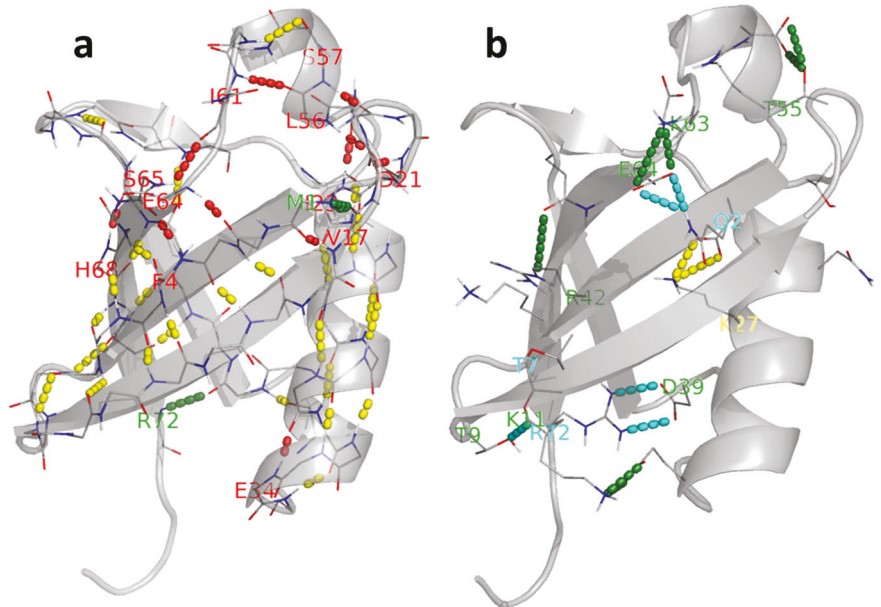

**Fig. 8 Hydrogen bonds in ubiquitin 3-D structure.** Only the H-bonds between (**a**) backbone atoms and (**b**) side chain atoms are displayed, where the structures are in transparent cartoon and the atoms involved in interactions are shown as wires. The H-bonds are represented as dashed lines and colored according to the sign of Pearson coefficients in Figs. 4, 7, where green for totally negative, red for totally positive, yellow for stable and pressure irrelevant, and cyan for unstable H-bonds in simulations. The donor residues were labeled in the same color scheme.

considerable occupancy include T55-D58 and K6-T12, and they did not have clear trend in response to pressure. In other words, H-bonds between side chains were rarely stabilized by pressure.

Those H-bonds with different Pearson correlations in ubiquitin structure were displayed in Fig. 8. The ones between backbones which maintain the helices and sheets are basically stable and less influenced by pressure. The pressure influenced hydrogen bonds are mainly formed between loops and the ends of helix and sheet. Half of them have large sequence separation (F4-S65, I23-R54, L56-D21, S57-P19, E64-Q2, H68-I44, and R72-Q40), coincident with what the previous NMR report supposed[29]. Most H-bonds between side chains were less stable and negatively influenced by pressure. For instances, during the simulations most systems lost

Q2-E64, T7–T9, and R72-D39, which were present in crystal structure.

The H-bonds formed between backbone N–H/C=O and side chain atoms were more stable and abundant than inter side-chain H-bonds. Among those with considerable average occupancy, E18-D21 (0.91), E51-Y59 (0.97), and D58-T55 (0.57) with amide H as donor had positive linear correlation with pressure, whereas S65-Q62 (0.87) and Q62-N60 (0.44) with carbonyl O as acceptor had negative correlation. The remaining H-bonds Q41-I36 (0.70), Q41-K27 (0.59), and K27-Q41 (0.55) formed a triad centered with Q41, and did not show homogenous correlation with pressure.

Some residues with backbone H-bond also interacted with each other through side chains, although only a few had considerable

occupancies (Table S3). The donor and acceptor of side chain H-bond E18-D21 were independent of the backbone H-bonds D21-E18, and both have positive correlation (Figs. 4, 7). Thus both H-bonds likely have synergized stabilization in response to pressure. For more conventional cases in which two H-bonds share the same backbone acceptor or donor, the correlations were opposite. A typical case is S65-Q62 where the backbone HN and side chain HG1 of S65 interacted with O of Q62 simultaneously. Originally HG1···O was stronger than HN···O, but as pressure increased the situation was reversed, because HN was pushed to O meanwhile HG1 was repulsed. This is an example that two H-bonds are competed with each other in response to pressure.

Compared to the H-bonds between backbones, the side chain H-bonds were less stable and easier to be destabilized by pressure. Although some side-chain involved H-bonds were constantly stable in all systems, in general circumstance the H-bond stability concerning with the composition in response to pressure can be ordered as, backbones > backbone—side chain > side chains. In other words, the H-bonds composed of side chains would bear the pressure destabilization in advance of the backbone H-bonds maintaining among helix and sheet.

As a feature of aqueously soluble protein, the water-bridged H-bonds are abundant in the simulations of ubiquitin, though most of them were insignificant for the low occupancies (Table S2). The water-bridged H-bond rarely occurs between backbones, and only L43–L50 and A46-T66 were constantly kept and all others required side chain atoms involved. More frequently they formed between the side chains with low sequence separation, such as the turn of loop and intra-helix. The mobility of water-bridged H-bonds was higher than direct H-bonds because the long-range movement of side chain makes larger possibility of H-bond rotations from one site to its neighbors, in the case that fast water exchange rate reduces the lifetime of H-bond events. As a consequence, since the occupancy variation of different systems is large, their linear correlations related to pressure are weaker than other H-bonds (Fig. 7b).

## Discussion
Inspired by available high-pressure NMR data for ubiquitin, we validated the transferability of the C36m protein force field with respect to pressure using MD simulations with LJ-PME method implemented. The $^{h3}J_{NC'}$ coupling is sensitive to identify the H-bond strength as its value is inverse to the exponential of HN···O distance (Eq. 1). Our MD simulations reproduced the $J$ couplings of backbone hydrogen bonds. To explore the relationship between pressure and protein structure from a range of discontinuous data, the regression is a useful way. According to the Pearson correlation coefficient, hydrogen bonds have three responses with respect to elevated pressure, enhanced ($\rho > 0$), destabilized ($\rho < 0$) and unclear ($\rho \approx 0$). The first two are straightforward, while the last one is more complicated because it may represent the case of either no correlation or nonlinear correlation. Due to the limited data points of pressure (6 or 10 in each temperature) by which the fitting of complicated functions is uncertain, we only consider the simple correlation in this study. For the strong H-bonds with high occupancy and small variance ($>0.7 \pm 0.1$) through all systems, low $|\rho|$ value means no correlation with pressure. On the other hand, for the H-bonds with large variance in occupancy, the correlation of low $|\rho|$ cannot be solved.

The pressure effect on backbone H-bonds was fully reproduced where the deviations of most reproduced $^{h3}J_{NC'}$ from NMR data are lower than 0.1 Hz. Large discrepancy systematically happened in seven sites, but five of them fit the H···O distance well referring to the crystal conformation. Admittedly the crystal structure is only comparable to the NMR measurement at 1 bar, and the

small difference does exist between two sets of experiments[38]. Combining both experiments as the reference, two H-bonds with constant deviation are I61-L56 (overstated) and T7-K11 (underestimated). I61-L56 is a loop H-bond across $3_{10}$-helix and T7-K11 is the first H-bond of β1/β2 hairpin, and both were influenced by the flexible neighbor residues in loops. Importantly such deviation is specific to certain H-bond pairs and is neither temperature nor pressure dependent. For this sake we may conclude C36m FF is capable of the simulations in a large range of $p$–$T$ conditions.

Sensitivity in response to the ambient pressure is different for secondary structure elements. As the linear regression of average RMSF shows, helices have negative coefficients under all temperatures other than sheets and loops (Table S1). The coefficients of determination ($R^2$) varied largely among temperature groups, and only the systems in 293 K show relatively good linear correlation for all domains. This is probably due to the insufficient data points of pressure. In 293 K, the RMSFs of helices, sheets and loops are equivalently decreased with respect to pressure. Furthermore the Pearson correlation coefficients of H-bonds between helices and loops are mostly positive, while only A28-E24 and N60-S57 showed considerable negative correlations. This suggests the H-bonds outside sheets are more easily to be compacted. Those observations are similar with that in NMR study on pancreatic tyrosine inhibitor, where the helices and loops have more compressibility than sheets[39].

The backbone H-bonds with negative linear correlation in all temperatures are M1-V17 ($N^+H_3$···O) between β1/β2 and R72-Q40 between β3/β5. They were presented as the end H-bonds from N- and C-terminal respectively. In NMR experiments, the $^{h3}J_{NC'}$ of both M1-V17 and R72-Q40 were not presented. Instead I3-L15 and R42-V70 are the first and last H-bonds with observed $^{h3}J_{NC'}$ values, respectively, and the strength of R42-V70 is negatively correlated with pressure[29,38]. According to the hypothesis from these NMR observations, the pressure denaturation of ubiquitin begins at position of R42-V70, the gate of C-terminal in β3/β5 strand. The phenomenon reproduced on R72-Q40 in simulations is largely consistent with R42-V70 in experiments, where the gate H-bond bears the destabilization but the neighbor is not influenced. The simulation results additionally indicate the pressure destabilization could also initiate on the gate H-bond of ionized N-terminal.

We also studied the pressure effects on side-chain involved H-bonds, which have not been systematically reported in experiments. Effective occupancy was computed for those H-bonds instead of the through-space $J$ couplings. Charged residues are the major contributors to the side-chain involved H-bonds, so the extra electrostatic attraction might enhance interaction energy between donor and acceptor. However, while most H-bonds between backbones were stabilized by elevated pressure, the H-bonds between side chains were mostly destabilized except for a few salt bridges which were not influenced. Similar to the observation from previous model studies using CHARMM parameters that Coulomb interaction energy of charged side chains is stronger in hydrophobic location than aqueous solvent[40,41], the exposed H-bonds between side chains are encountered more water competitions than buried backbone H-bonds. Therefore, the observation that the stability of side chain H-bonds in response to pressure is not comparable to backbone ones is expected, due to the water dynamics increased by elevated pressure.

The flexibilities of direct and water-bridged H-bonds are different in response to pressure. The lifetime of H-bond events varied from several picoseconds to hundreds of nanoseconds in long MD simulation. For direct H-bonds, the shorter events (lifetime ≤6 ns) are decreased along with the elevation of pressure, while the longer events (6 ns < lifetime ≤ 100 ns) are increased

(Fig. S8). This is reasonable that the breath frequency of some H-bonds is reduced as the distances between donor and acceptor are compressed. The exception is for a few constantly long events (lifetime > 100 ns), which are originally rigid and not influenced by the pressure. In general pressure reduces the dynamics of global inter-residue H-bonds and elongates the average lifetimes of the interaction. In contrast, although the number of water-bridged H-bonds is significantly larger, most of these H-bonds are transient as lifetimes being shorter than 4 ns. Those short events are increased with respect to the rising of hydrostatic pressure. Compared to direct H-bonds, more H-bonds had been formed between residues and water molecules but few got reduced breath frequency. This is consistent with the observation of another computational work for short peptide and TIP4P water model, where the H-bonds between backbone and solvent are more dynamic in high pressure at 300 K thereby producing more short events[42]. Elevated pressures increase the probability of H-bonds between water and protein surface residues by compressing their atomic radii. Yet at 2500 bar, there is still no extra cavity caused by water penetration according to the hypothesis of pressure denaturation proposed in another MD simulation[43].

Water-bridged contact between hydrophobic side-chain depicts the non-polar interaction between solvent and protein. Totally 24 water-bridged contacts were observed and they are only located on solvent exposed residues, i.e., no water was observed between the side chains of the 14 residues that is defined as hydrophobic core. This implies water penetration does not happen when the volume of hydrophobic core compresses, in agreement with the NMR conclusion that up to 2500 bar ubiquitin is stable[26,29]. Similar to H-bonds, the events of water-bridged contact were tremendously increased with pressure as the distance between water and hydrophobic side chains were compacted. Such non-polar compression has stronger linear correlation with pressure since the enhancement of hydrophobic contacts is directionless compared to H-bonds (Fig. 7c). In summary both water-bridged H-bonds and contacts confirm the access radius of surface atoms is reduced in response to hydrostatic pressure while maintaining the conformation.

Several computational studies have studied the water penetration into the ubiquitin hydrophobic core as the ambient pressure increased up to 3000 bar using Amber94[43] based on the ensemble in NMR study[44], and up to 10000 bar using CHARMM22 with a random walk sampling in pressure space[45]. By analyzing the energetic profile of solvation shell, those authors concluded that the events of water entering hydrophobic core are the most important transition states for pressure induced structural deformation. Indeed the infrared spectroscopy shows that the ubiquitin unfolding occurs at 5400 bar, where the secondary structures were rearranged rather than disordered[46]. The prediction of pressure destabilization that extends the phase diagram from native to denatured state is out of the scope in this study. Instead, by reproducing the experimental H-bonds in response to hydrostatic pressure we identified the ubiquitin is stable up to 2500 bar, but the polar and non-polar interactions in transition states have different responses during the compression. These effects can be considered when interpreting pressure denaturation in other studies.

## Conclusion

The major purpose of this study is to evaluate the capability of CHARMM36m additive force field combined with the LJ-PME nonbonded method in a wide range of temperature and pressure. By reproducing the backbone hydrogen bonds and the internal motion of methyl groups from 1 to 2500 bar based on long MD simulations, the calculated properties are in good agreement with NMR observations. The deviations are independent of the temperature and pressure of the simulation systems. These results verified that protein force field CHARMM36m has good transferability in MD simulations with different conditions of pressure.

Following the experimental data, the study also explored the transition states of ubiquitin under the elevated pressure. The pressure generally reduces the residue flexibility and helices are more sensitive than sheets and loops. Except for gate H-bonds between sheets whose strength is weakened, most H-bonds between backbones are either enhanced or not influenced in response to pressure. On the other hand, the side chain involved H-bonds are more fragile to bear the pressure destabilization, which indicates that the pressure denaturation starts from the side-chain H-bonds and then the network between sheets. Pressure reduces fluctuation of inter-residue H-bonds meanwhile increases both the hydrophilic and hydrophobic contacts by compressing the vdW radii. However up to 2500 bar, no cavity is formed to enable water penetration in ubiquitin. The feasibility of modern force fields combined with LJ-PME would encourage more interest in computational study on biomolecules under elevated pressures.

## Methods

**MD simulations**. The ubiquitin structure (PDB ID: 1UBQ[47], 1.8 Å) was used as the model system in different thermodynamic conditions. All the titratable residues were treated as ionized and histidine 68 was protonated on $\delta$-N. It was solvated in a TIP3P water[48] box with 58.5 Å as the cubic dimension. The N- and C-terminal were capped by standard ammonium and carbonate forms respectively and no counter-ions and salt concentration were added. The system totally has 20290 atoms. Other proteins which were investigated for validation of LJ-PME simulations include cold-shock protein A (1MJC), protein G B1 domain (2QMT, 1.1 Å), apo-calmodulin (1QX5), and intestinal fatty acid binding protein (1IFC). The setup and pre-equilibrium of these systems were done following the same procedure in a previous study[20].

Periodic boundary conditions were used and the PME[49] were applied on both electrostatic and Lennard-Jones (LJ-PME) interactions, with 9 Å as the real space cutoff and $10^{-4}$ as the error tolerance. The simulations were performed in $NpT$ ensemble by employing Andersen thermostat and Monte Carlo barostat. For 1UBQ the pressure and temperature conditions were designed according to the NMR experimental study[29] with a few more simulations added to make the analysis more systematic (Table 2). Simulations of all other proteins were carried out only at room condition (300 K and 1 bar). C36m[24,50] FF and the CHARMM modified TIP3P water model were applied. With the constraint on hydrogen involved bonds, a time step of 2 fs was used in velocity Verlet integrator. Each system was run 1.2 μs using OpenMM 7[51] with snapshot saving in every 100 ps.

**Analysis**. The last 1 μs from each 1.2 μs trajectory including 10,000 snapshots was accounted in analysis using CHARMM[52]. The strength of hydrogen bond quantified using the scalar coupling between the backbone N and C atoms was considered as the fingerprint to identify the structural stability. The scalar coupling is calculated using the following formula as described previously[53],

$$^{h3}J_{NC'} \ = \ \langle (-357\,\text{Hz}) \exp(-3.2 r_{HO}) \cos^2 \theta \rangle \tag{1}$$

where $r_{HO}$ is the distance between the hydrogen and the acceptor oxygen atom, $\theta$ is the H···O=C angle, and angular bracket stands for ensemble average over the MD trajectories.

The global intramolecular hydrogen bonds in ubiquitin were identified during MD simulations considering the criteria that $r_{HO} \leq 2.4$ Å and $\theta \geq 100°$. For a residue which has multiple equivalent donors or acceptors involved in an identical interaction during the same lifetime span, these events were degenerated to one. The percentage of degenerated dwell time over whole MD trajectory is effective occupancy, and it was calculated for all possible H-bonds. The equivalent donors or acceptors mean the following atoms of residues: Asn (HD21, HD22), Gln (HE21, HE22), Lys (HZ1, HZ2, HZ3), Arg (HE, HH11, HH12, HH21, HH22), N-terminal (HT1, HT2, HT3), Asp (OD1, OD2), Glu (OE1, OE2), and C-terminal (OT1, OT2). The effective occupancy of water-bridged hydrogen bonds where a water molecule sits between two polar residues was also considered in this way. Similar to H-bonds, the effective hydrophobic contact was counted for the distance between the mass centers of two hydrophobic side chains. The cutoff was 3.5 Å between any pair of mass centers and no direction criterion was set.

The order parameter $S^2$ of methyl group which can be directly related to the NMR relaxation data was calculated using the trace of tensor and taking the

ensemble average, as described before[28],

$$S^2 = \frac{3}{2}\text{tr}\langle\Phi\rangle^2 - \frac{1}{2}(\text{tr}\langle\Phi\rangle)^2 \qquad (2)$$

where $\Phi$ is a $3 \times 3$ tensor of a unit vector along the C–C axis of side chain methyl group for each specified residue, i.e.,

$$\Phi_{ij} = \frac{r_i r_j}{r^2} \qquad (3)$$

where $r_i$ ($i = x, y, z$) is the coordinate component for vector $r$ ($r^2 = r_x^2 + r_y^2 + r_z^2$) of the methyl carbon related to the attached carbon in protein. In practice, the vectors were computed for each snapshot and averaged over the whole trajectory as the ensemble average.

## Data availability

The CHARMM topology (ubi.psf) and coordinate (ubi.crd) and the input of OpenMM (run.py) which can reproduce the systems in different $p$–$T$ conditions were pushed on GitHub (https://github.com/youxu0/ubiquitin-inputs).

## Code availability

Scripts and codes are made available on GitHub (https://github.com/youxu0/ubiquitin-inputs).

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

## Acknowledgements
The work is supported by Zhejiang Provincial Natural Science Foundation of China (Grant No. LR19B030001 and LQ20C050001), National Natural Science Foundation of China (Grant No. 21803057), Westlake Education Foundation and Tencent Foundation. We thank Westlake University Supercomputer Center for computational resource and related assistance.

## Author contributions
J.H. initiated the project, designed the experiment, and wrote the paper. Y.X. performed the simulations, analyzed the data, and wrote the paper.

## Competing interests
The authors declare no competing interests.
