## [Peer Review File · Communications Chemistry]

Reviewers' comments:

Reviewer #1 (Remarks to the Author):

This is an interesting paper that reports an MD study of the behavior of ubiquitin at various pressure and temperature conditions. The Authors' main objective was to assess the performance of the CHARMM36m force field with Ewald-like treatment of the nonbonded potential. They calculated the coupling constants and order parameters and compared these values and their variation with temperature and pressure with the experimental counterparts, finding good agreement. Overall, the paper is interesting and the study has been performed and the results described well. I recommend publication subject to a minor revision.

Apart from the assessment, an interesting part of the paper is the discussion of the change of hydrogen-bond and hydrophobic-contact strength with pressure. Some of the hydrogen-bonding/hydrophobic contacts are tightened and some are loosened. However, the Authors only describe the facts, while they seem to be reluctant in interpreting. For example, the two sentences in lines 442-445:

"However, while most H-bonds between backbones were stabilized by elevated pressure, the H-bonds between side chains were mostly destabilized except for a few salt-bridges which were not influenced. The results that backbones involved H-bonds have better stabilization in response to pressure than side chains suggest the dynamic of side chain should be a more dominant factor than charge attraction to the maintenance of H-bonds in simulations."

The dynamics can be one factor that determines stabilization but increasing pressure should also affect negatively the stability of surface hydrogen-bonding contacts compared to backbone hydrogen-bonding contacts that are shielded from the solvent. In this regard, it would be useful to lookup the literature for model studies such as, e.g., those of Lazaridis or Makowski of pairs of amino-acid charged side chains in water.

Further to the above, I'd suggest making a figure with a structure of ubiquitin and hydrogen/hydrophobic contacts, indicating which ones are stabilized and which are destabilized by elevated pressure (e.g., color them according to the value of the Pearson coefficient). Such a figure could illustrate the origin of stabilization/destabilization of these contacts. A plot of the Pearson coefficients as a function of solvent accessible surface area of the atoms involved in hydrogen bonds would also be useful to assess how much is the pressure effect on hydrogen-bonding constants dependent on the exposure of to solvent.

Minor points:

1. Line 154: The definition of Φ is not complete. It is not clear how the tensor is calculated. The respective formula should appear.
2. Lines 192-192: "As RMSD of the full structure is hardly to differentiate the local conformation," The Authors probably mean "RMSD... is insufficient to differentiate".
3. Lines 483-485: "Their simulations suppose the mechanism of pressure denaturation through the energetic

profile of solvation shell, in which the water entering hydrophobic core is the most important transition state for structural deformation."

This sentence is unclear, seems to have grammar issues.

Reviewer #2 (Remarks to the Author):

The Authors of the Manuscript "On the Transferability of CHARMM36m Protein Force Field with LJ-PME: Hydrogen Bonding Dynamics under Elevated Pressures" have provided a much needed benchmark for the commonly used CHARMM36m force field extended to high pressures. At the same time, Authors evaluate the LJ-PME method for use with this force field. I like the very thorough analysis performed by the Authors and meticulous comparison to the high pressure NMR experiments. I believe the work is an important sanity check, which turns out very favourably for the CHARMM36m force field. It will likely have a large influence on the field of the high-pressure simulations and should be published.

Before I can wholeheartedly recommend publication of the Manuscript in Communications Chemistry, I would like to see two points of criticism addressed:

1. The Authors show the RMSD dependence on the pressure (Fig 1.). I find this Figure slightly confusing, in particular the jumps in RMSD of the lowest and highest temperature systems. Did Authors detect the source of these changes? Was there any large conformational change observed? If so, it could explain the changes in the RMSF profiles (e.g. the dip in the RMSF for beta sheets and helices for highest T simulation in Fig 2). It could potentially mask additional pressure-related effects. I would find the results much more convincing if the Authors provided an additional set of 2 simulations for at least one of the conditions I mentioned (e.g. P=2000 and T=323 or P=1000 and T=278). Would this affect the RMSF behaviour or calculated J-couplings?

2. The calculation of the cavity. Authors used the correlation of the distances of the atoms within the hydrophobic pocket as an indication of the compression. It is however difficult to estimate the actual size of the cavity and degree of the change upon pressure elevation. A more direct way of calculating the cavity volume would allow for a direct comparison with the known high-pressure structures of the ubiquitin. Would a simple calculation with Caver or the new method by Chwastyk et al.: <https://pubmed.ncbi.nlm.nih.gov/27231838/> show the same trend? Could it be compared to the existing structures?

3. To facilitate the reproducibility of the study, the files needed to re-run the simulations should be made available. At least the equilibrated systems together with input files should be made available in a public repository.

Besides that, the abbreviation H-bond is used without introduction.

Reviewer #3 (Remarks to the Author):

The study titled: "On the transferability of charmm36m protein force field with LJ-PME: Hydrogen

bonding dynamics under elevated pressures" led by professor J. Huang shows how a charmm36m force field in combination with the LJ-PME method fit correctly with previously published experimental data (J couplings for hydrogen bonds and "S" parameters) into the ubiquitin protein system (1UBQ). It is an article with a high level of detail and it's easily followed. The Figure S5 shows us with a high sincerity on the part of the authors how the force fields adjusts to the experimental results. I have read this interesting work in detail and care, and I really find the results novel and sound. However, I also think that the validation of a force field cannot only be studied with a single system of interest and a more extensive study with more protein systems are necessary to know the benefits of this force field with pressure. I have the feeling that this work could be better located in a more specialized journal in theoretical chemistry too.

We kindly appreciate the critical comments from the reviewers. Here are our response and feedback after we thoroughly took account of all the issues that reviewers have pointed out.

Reviewer 1

General comments: *This is an interesting paper that reports an MD study of the behavior of ubiquitin at various pressure and temperature conditions. The Authors' main objective was to assess the performance of the CHARMM36m force field with Ewald-like treatment of the nonbonded potential. They calculated the coupling constants and order parameters and compared these values and their variation with temperature and pressure with the experimental counterparts, finding good agreement. Overall, the paper is interesting and the study has been performed and the results described well. I recommend publication subject to a minor revision.*

We would like to thank the reviewer for his or her careful reading and positive comments.

Q1. *Apart from the assessment, an interesting part of the paper is the discussion of the change of hydrogen-bond and hydrophobic-contact strength with pressure. Some of the hydrogen-bonding/hydrophobic contacts are tightened and some are loosened. However, the Authors only describe the facts, while they seem to be reluctant in interpreting. For example, the two sentences in lines 442-445:*

"However, while most H-bonds between backbones were stabilized by elevated pressure, the H-bonds between side chains were mostly destabilized except for a few salt-bridges which were not influenced. The results that backbones involved H-bonds have better stabilization in response to pressure than side chains suggest the dynamic of side chain should be a more dominant factor than charge attraction to the maintenance of H-bonds in simulations."

The dynamics can be one factor that determines stabilization but increasing pressure should also affect negatively the stability of surface hydrogen-bonding contacts compared to backbone hydrogen-bonding contacts that are shielded from the solvent. In this regard, it would be useful to lookup the literature for model studies such as, e.g., those of Lazaridis or Makowski of pairs of amino-acid charged side chains in water.

A1. We agree with the reviewer. The H-bond is always weakened in aqueous solvent compared to buried environment due to the competition from water molecules. The pressure makes the water exchange more frequent thus decreasing the stability of exposed H-bond further. We added the following sentences in the discussion section with two related papers of Lazaridis cited.

"Similar to the observation from previous model studies using CHARMM parameters that Coulomb interaction energy of charged side chains is stronger in hydrophobic location than aqueous solvent, the exposed H-bonds between side-chains are encountered more water competitions than buried backbone H-bonds. Therefore, the observation that the stability of side chain H-bonds in response to pressure is not comparable to backbone ones is expected, due to the water dynamics increased by elevated pressure." (page 13)

Q2. *Further to the above, I'd suggest making a figure with a structure of ubiquitin and hydrogen/hydrophobic contacts, indicating which ones are stabilized and which are destabilized by elevated pressure (e.g., color them according to the value of the Pearson coefficient). Such a figure could illustrate the origin of stabilization/destabilization of these contacts. A plot of the Pearson coefficients as a function of solvent accessible surface area of the atoms involved in hydrogen bonds would also be useful to assess how much is the pressure effect on hydrogen-bonding constants dependent on the exposure of to solvent.*

A2. We thank the reviewer's suggestion and added a set of new plots (Figure 8) clustering the H-bonds between backbones and side chains respectively according to the Pearson correlation. They show the hydrogen bonds in 3D structure of ubiquitin and colored based on the sign of Pearson coefficients. Most H-bonds between backbones are stable and not influenced by pressure, while the destabilized ones are the gate on sheets and the stabilized ones are mainly located between loops and the ends of helix or sheets. The H-bonds between side chains however were mostly destabilized by pressure. A paragraph was added in the Results section following the presentation of H-bonds between side chains to discuss this.

"Those H-bonds with different Pearson correlations in ubiquitin structure was displayed in Figure 8. The ones between backbones which maintain the helices and sheets are basically stable and less influenced by pressure. The pressure influenced hydrogen bonds are mainly formed between loops and the ends of helix and sheet. Half of them have large sequence separation (F4-S65, I23-R54, L56-D21, S57-P19, E64-Q2, H68-I44 and R72-Q40), coincident with what the previous NMR report supposed. Most H-bonds between side chains were less stable and negatively influenced by pressure. For instances, during the simulations most systems lost Q2-E64, T7-T9 and R72-D39, which were present in crystal structure." (page 11)

We also calculated the solvent accessible surface area (SASA) for each atom which is participating H-bonds using GETAREA (J Comput Chem 19: 319–333, 1998). Most backbone N atoms (except for residues 17 and 35) and O atoms (except for residues 6, 28, 31, 45 and 57) are buried and no correlation between surface and pressure was shown for those residues. Although the side chain atoms have more considerable surface area, we didn't find out any informative relationship between the area and pressure. Since the conformational change is very limited among different ubiquitin systems, we think SASA might not be a significant feature to identify the hydrogen bond properties in response to pressure.

Q3. Minor points:

1. Line 154: *The definition of Φ is not complete. It is not clear how the tensor is calculated. The respective formula should appear.*

2. Lines 192-192: *"As RMSD of the full structure is hardly to differentiate the local conformation," The Authors probably mean "RMSD... is insufficient to differentiate".*

3. Lines 483-485: *"Their simulations suppose the mechanism of pressure denaturation through the energetic profile of solvation shell, in which the water entering hydrophobic core is the most*

important transition state for structural deformation." This sentence is unclear, seems to have grammar issues.

A3. We thank the reviewer for his or her careful reading. The definition of Φ is added.

"where Φ is a 3×3 tensor of a unit vector along the C-C axis of side chain methyl group for each specified residue, *i.e.*,

$$\Phi_{ij} = \frac{r_i r_j}{r^2} \quad (1)$$

where r_i ($i = x, y, z$) is the coordinate component for vector r ($r^2 = r_x^2 + r_y^2 + r_z^2$) of the methyl carbon related to the attached carbon in protein." (page 6)

We also rephrased the two sentences that the reviewer kindly pointed out.

"As RMSD of the full structure is insufficient to differentiate the local conformation," (page 7)

"By analyzing the energetic profile of solvation shell, those authors concluded that the water entering hydrophobic core is the most important transition state for pressure induced structural deformation." (page 14)

Reviewer 2

General comments: *The Authors of the Manuscript "On the Transferability of CHARMM36m Protein Force Field with LJ-PME: Hydrogen Bonding Dynamics under Elevated Pressures" have provided a much needed benchmark for the commonly used CHARMM36m force field extended to high pressures. At the same time, Authors evaluate the LJ-PME method for use with this force field. I like the very thorough analysis performed by the Authors and meticulous comparison to the high pressure NMR experiments. I believe the work is an important sanity check, which turns out very favourably for the CHARMM36m force field. It will likely have a large influence on the field of the high-pressure simulations and should be published.*

We would like to thank the reviewer for his or her careful reading and favorable comments.

Q1. *Before I can wholeheartedly recommended publication of the Manuscript in Communications Chemistry, I would like to see two points of criticism addressed:*

The Authors show the RMSD dependence on the pressure (Fig 1.). I find this Figure slightly confusing, in particular the jumps in RMSD of the lowest and highest temperature systems. Did Authors detect the source of these changes? Was there any large conformational change observed? If so, it could explain the changes in the RMSF profiles (e.g. the dip in the RMSF for beta sheets and helices for highest T simulation in Fig 2). It could potentially mask additional pressure-related effects. I would find the results much more convincing if the Authors provided an additional set of 2 simulations for at least one of the conditions I mentioned (e.g. P=2000 and T=323 or P=1000 and T=278). Would this affect the RMSF behaviour or calculated J-couplings?

A1. We thank the reviewer's suggestion and have checked the RMSD and RMSF calculations. The jump of group 1000 bar/278 K was caused by the mobile tail residue 72 which was unstable in many systems. As we discussed in text, R72-Q40 is the final H-bond between $\beta 5$ and $\beta 5$ strand and unstable in simulations. However H-bond R72-Q40 was not observed in NMR experiments probably due to its mobility. So we think it is better to exclude the flexible loop 72-76 and compute the RMSD for only residues 1-71. The same exclusion is also applied for RMSF calculations. Accordingly, Table 1, Figure 1 and Figure 2 were updated as well as with Figure S3. The values of both RMSD and RMSF are therefore reduced a little and the tendency in Table 1 and Figure 2 becomes more concise.

The peak at 2000 bar/323 K however was not caused by residue 72. Instead, the high RMSD was contributed by the random fluctuation of helix 1 and loop 2 (Figure A1, the time series of RMSD). Those destabilization events are frequently happened in the mobile region loop 2 and $\beta 3$ - $\beta 4$ (hairpin 2) with some time span. To reproduce this system, we ran four additional 1.2 μ s simulations under the condition of $p = 2000$ bar and $T = 323$ K as the reviewer suggested. Two of these simulations showed higher average RMSD (2.35 and 2.10 \AA) and the other two have lower average RMSD (1.73 and 1.72 \AA) compared to the original 1.89 \AA . The destabilization from first two runs was mainly caused by loop 2 and hairpin 2: they both lose the C-terminal gate H-bond R72-Q40 and one simulation further loses the next H-bond R42-V70 and the commonly flexible T7-K11 in hairpin 1 (Figure A2).

In the manuscript we have discussed the flexibility of T7-K11 and destabilization of R72-Q40 in response to elevated pressure, on which structural fluctuations were randomly happened in several systems. As comparisons, two of the four newly performed MD simulations are more stable and no significant events were observed on those domains. Thus the RMSD would look more reasonable if the data in Figure 1 was replaced by either of them. Despite of that, the distances of the remaining H-bonds are almost not diverse among all replications (Figure A3), therefore the J -coupling profiles of these H-bonds are essentially the same for these simulations.

Providing that, we still keep the original data presentation of the 2000 bar/323 K system, and the presentation described in the manuscript of this part is unchanged.

Figure A1. RMSDs with respect to time for local domains.

Figure A2. The RMSF for residues 1—71 of five simulations at $p = 2000$ bar and $T = 323$ K. Color scheme: blue for the original one presented in the manuscript, and marine, green, brown and red for newly performed simulations.

Figure A3. The distance between donor and acceptor of backbone H-bonds of five simulations at $p = 2000$ bar and $T = 323$ K. Same color scheme was used as Figure A2.

Q2. *The calculation of the cavity. Authors used the correlation of the distances of the atoms within the hydrophobic pocket as an indication of the compression. It is however difficult to estimate the actual size of the cavity and degree of the change upon pressure elevation. A more direct way of calculating the cavity volume would allow for a direct comparison with the known high-pressure structures of the ubiquitin. Would a simple calculation with Caver or the new method by Chwastyk et al.: <https://pubmed.ncbi.nlm.nih.gov/27231838/> show the same trend? Could it be compared to the existing structures?*

A2. We have tried both methods (CAVER and SPACEBALL) to calculate the cavity of our systems. The webserver of SPACEBALL however seems to be in hibernation, as we are not able to get results after job submission. The tunnel information calculated by CAVER shows the shrinkage of the cavity volume in response to pressure, which is consistent with the distance-pressure correlation of hydrophobic side chain in our manuscript. We added a new figure (Figure S7) in the supporting information, and added the following description in the main text.

“The cavity volume of ubiquitin was evaluated for the systems at 308 K using CAVER3.0⁴⁴ (Figure S7). Tunnels were vanished gradually as bottleneck radius was decreased, which confirms that the cavities in ubiquitin are compacted by pressure.” (page 10)

Q3. To facilitate the reproducibility of the study, the files needed to re-run the simulations should be made available. At least the equilibrated systems together with input files should be made available in a public repository.

A3. We have deposited the corresponding CHARMM topology (ubi.psf) and coordinate (ubi.crd) and the python script of running OpenMM (run.py) which can reproduce the systems in different p - T conditions in GitHub (<https://github.com/youxu0/ubiquitin-inputs>). We have also added a subsection “Data availability” to mention this.

Q4. *The abbreviation H-bond is used without introduction.*

A4. The introduction has been made in its first appearance in the main text (page 3).

Reviewer 3

General comments: *The study titled: “On the transferability of charmm36m protein force field with LJ-PME: Hydrogen bonding dynamics under elevated pressures” led by professor J. Huang shows how a charmm36m force field in combination with the LJ-PME method fit correctly with previously published experimental data (J couplings for hydrogen bonds and “S” parameters) into the ubiquitin protein system (IUBQ). It is an article with a high level of detail and it’s easily followed. The Figure S5 shows us with a high sincerity on the part of the authors how the force fields adjusts to the experimental results. I have read this interesting work in detail and care, and I really find the results novel and sound. However, I also think that the validation of a force field cannot only be studied with a single system of interest and a more extensive study with more protein systems are necessary to know the benefits of this force field with pressure. I have the feeling that this work could be better located in a more specialized journal in theoretical chemistry too.*

We thank the reviewer for his or her favorable comments on this work and are very happy that the reviewer found our results novel and sound. We believe that this work provides both new insights and useful information in studying protein dynamics under high-pressure, and *Communications Chemistry* would be an ideal venue for it to reach a broad audience.

REVIEWERS' COMMENTS:

Reviewer #1 (Remarks to the Author):

The Authors have addressed my comments sufficiently and I now can recommend the manuscript for publication. I only found one minor grammar error in the added text in page 11:

"Those H-bonds with different Pearson correlations in ubiquitin structure was displayed in Figure 8."

The second line should probably read "...were displayed..."

Reviewer #2 (Remarks to the Author):

I feel the Authors have addressed all the remarks and am fully supporting the publication.

Reviewer #1

Q1. I only found one minor grammar error in the added text in page 11:

"Those H-bonds with different Pearson correlations in ubiquitin structure was displayed in Figure 8." The second line should probably read "...were displayed..."

A1. Thanks for pointing this out. We have corrected it.